# Disruptions in Effective Connectivity within and between Default Mode Network and Anterior Forebrain Mesocircuit in Prolonged Disorders of Consciousness

**DOI:** 10.3390/brainsci11060749

**Published:** 2021-06-04

**Authors:** Sean Coulborn, Chris Taylor, Lorina Naci, Adrian M. Owen, Davinia Fernández-Espejo

**Affiliations:** 1Centre for Human Brain Health and School of Psychology, University of Birmingham, Birmingham B15 2TT, UK; sic613@student.bham.ac.uk (S.C.); chris.taylor.20@ucl.ac.uk (C.T.); 2Trinity College Institute of Neuroscience, School of Psychology, Trinity College Dublin, D02 PN40 Dublin, Ireland; nacil@tcd.ie; 3Brain and Mind Institute, Western University, London, ON N6A 5B7, Canada; aowen6@uwo.ca

**Keywords:** disorders of consciousness, default mode network, anterior forebrain mesocircuit, dynamic causal modelling, effective connectivity, fMRI, parametric empirical Bayes, vegetative state, minimally conscious state, coma recovery scale-revised

## Abstract

Recent research indicates prolonged disorders of consciousness (PDOC) result from structural and functional impairments to key cortical and subcortical networks, including the default mode network (DMN) and the anterior forebrain mesocircuit (AFM). However, the specific mechanisms which underpin such impairments remain unknown. It is known that disruptions in the striatal-pallidal pathway can result in the over inhibition of the thalamus and lack of excitation to the cortex that characterizes PDOC. Here, we used spectral dynamic causal modelling and parametric empirical Bayes on rs-fMRI data to assess whether DMN changes in PDOC are caused by disruptions in the AFM. PDOC patients displayed overall reduced coupling within the AFM, and specifically, decreased self-inhibition of the striatum, paired with reduced coupling from striatum to thalamus. This led to loss of inhibition from AFM to DMN, mostly driven by posterior areas including the precuneus and inferior parietal cortex. In turn, the DMN showed disruptions in self-inhibition of the precuneus and medial prefrontal cortex. Our results provide support for the anterior mesocircuit model at the subcortical level but highlight an inhibitory role for the AFM over the DMN, which is disrupted in PDOC.

## 1. Introduction

Recent years have seen an increasing interest in characterizing the neural bases underlying the (full or partial) lack of awareness in prolonged disorders of consciousness (PDOC) with neuroimaging techniques [1]. Alongside contributing to advancing our understanding of the neural correlates of consciousness [2], the outcomes of this research can provide biomarkers to improve diagnosis and prognostication accuracy, as well as assist with the development of potential therapies. Much of the work to date has focused on structural and (resting-state) functional connectivity [3]. This has identified disconnections in long-range fronto-parietal networks, and particularly the default mode network (DMN) [4,5,6,7,8,9,10,11,12,13], as a key contributor to the lack of awareness in PDOC. The DMN is an intrinsic brain network encompassing the posterior cingulate cortex/precuneus (PCC), bilateral inferior parietal lobules (IPL), and the medial prefrontal cortex (MPFC), which show synchronic activity in individuals at rest [14]. There is increasing evidence of structural [5,7,8] and functional disconnections [9,10,11,12,13] as well as metabolic impairments in the DMN in PDOC [15,16], the severity of which correlates with clinical severity [7,8,10,11,12,13,16] and prognosis [7,17]. For example, recently, intact functional connectivity of the DMN was associated with later recovery (observed through command following) in a patient who became unresponsive after contracting COVID-19 [18]. The DMN has a key role in the generation of self-referential processes and awareness [4,19], suggesting its deficits could specifically explain the lack of self-awareness in PDOC.

In parallel, the so called mesocircuit hypothesis [20] proposes that the lack of awareness in PDOC is caused by disbalances in the anterior forebrain mesocircuit (AFM) that result in an excess of thalamic inhibition and lack of excitation to the cortex. In support of this model, positron emission topography (PET) studies have confirmed reduced metabolic uptake in the striatum, thalamus, and fronto-parietal regions, coupled with increased metabolism of the globus pallidus in PDOC [16]. According to the mesocircuit model, the striatum fails to inhibit the globus pallidus, which becomes hyperreactive and excessively inhibits the thalamus. This excess of inhibition in an already damaged and hypoactive thalamus [21] in turn leads to severely reduced excitatory output to high order frontoparietal regions. The mesocircuit model provides a framework to interpret causal mechanisms explaining the, sometimes paradoxical, positive effect of pharmacological agents such as zolpidem or amantadine on the basis of their effects on striatal and thalamic populations [22,23,24,25].

We have previously demonstrated that these models are not mutually exclusive. In fact, the DMN and AFM are highly interconnected via well-established thalamo-cortical and corticostriatal white matter paths, and these show specific damage that correlates with severity in PDOC [8]. In line with earlier work focusing on the DMN alone [5], we identified a key role for the PCC in the relationship between DMN and AFM. Specifically, we confirmed structural damage to the white matter fibers connecting the PCC with all major nodes in the DMN, as well as with key subcortical structures including the thalamus and striatum [8]. The PCC is known to be highly structurally connected to other high order regions in the brain and is considered to be one of the major hubs for the functional integration across networks [26,27,28]. In line with our structural findings, dynamic causal modelling (DCM) provided support to the PCC’s role in mediating functional disconnections in the DMN in PDOC [6]. Similarly, a recent study demonstrated abnormal effective connectivity in anterior forebrain regions in PDOC [29]. Here, we expand these results by using DCM to investigate causal dynamics within and between regions of the DMN and AFM in a group of PDOC patients, including both diagnoses of vegetative and minimally conscious state. We aim to confirm previously reported disconnections and disbalances in both networks but also to establish whether (1) cortical changes in the DMN are caused by lack of excitation from the thalamus (as predicted by the mesocircuit model) and (2) this is modulated by the PCC specifically (as earlier structural and functional research suggests).

## 2. Materials and Methods

### 2.1. Participants

We recruited a convenience sample of 18 PDOC patients, in a vegetative or minimally conscious state, at the University of Western Ontario (UWO) between 2012 and 2015. Exclusion criteria included age under 18 years, and lack of eligibility to enter the MRI environment. Two patients were discarded due to poor data quality (excessive movement in the scanner > 3 mm translation and 3° radius for more the 15% of the acquired scans) leaving a final sample of 16 patients (7 female, aged 20–56, M = 35.94, SD = 11.91). Patients were behaviorally assessed repeatedly through administrations of the Coma Recovery Scale-Revised (CRS-R) [30] over a 5-day period during the week they were scanned (Median number of CRS-R assessments = 3, range = 1–5). Independent functional [31,32,33,34,35] and structural [8,36] subsets of this dataset have previously been reported. Table 1 contains clinical details of each patient.

We also recruited 16 right-handed healthy volunteers (7 female, aged 19–29, M = 25.43, SD = 2.53), with no history of psychiatric or neurological disorders, between 2013 and 2014.

The UWO’s Health Sciences Research Ethics Board (London, Ontario, Canada) provided ethical approval for the study. All healthy volunteers provided informed written consent. For each patient, a surrogate decision maker provided informed written consent to participate in the study.

### 2.2. MRI Acquisition

We acquired MRI data with a 3T Siemens scanner (Siemens, Erlangen, Germany) and a 32-channel head-coil at the Centre for Functional and Metabolic Mapping at Robarts Research Institute (London, Canada). The patients were recruited over a 3-year period, during which time the 3T scanner was upgraded. Seven patients and 7 healthy volunteers were scanned before the upgrade, in a Magnetom Trio system, and the remaining 9 patients and 9 healthy volunteers were scanned in the new Magnetom Prisma system. This resulted in a balanced distribution of patients and healthy controls across the two scanners. The fMRI acquisition consisted of functional echo-planar images of 36 slices covering the whole brain with the following parameters: TR  =  2000 ms, TE  =  30 ms, matrix size  = 70 × 70, slice thickness  =  3 mm, in-plane resolution  =  3 × 3 mm, flip angle  =  78°, 245 volumes (9 healthy controls and 10 patients), and TR = 2000 ms, TE = 30 ms, matrix size = 64 × 64, slice thickness  =  3 mm, in-plane resolution  =  3 × 3 mm, flip angle = 75°, 150 volumes (7 healthy controls and 6 patients).

In addition to the fMRI data, we acquired a high-resolution, T1-weighted, 3-dimensional magnetization prepared rapid acquisition gradient echo (MPRAGE) image during the same session (Trio system: TR = 2300 ms, TE = 2.98 ms, inversion time = 900 ms, matrix size = 256 × 240, voxel size = 1 × 1 × 1 mm, flip angle = 9°; Prisma system: TR = 2300 ms, TE = 2.32 ms, inversion time = 900 ms, matrix size = 256 × 256, voxel size = 1 × 1 × 1 mm, flip angle = 8°; for five patients matrix size = 240 × 256 and flip angle = 9°).

### 2.3. Preprocessing

We reprocessed all data using SPM12 (http://www.fil.ion.ucl.ac.uk/spm accessed on 1 April 2021) running on MATLAB version R2014b. We first reoriented the data according to the AC-PC. Spatial pre-processing included: realignment to correct subjects’ motion, co-registration between the functional and structural data sets, and smoothing with an 8 mm full width half maximum Gaussian kernel. Finally, to reduce noise, we applied a bandpass temporal filtering between 0.009–0.08 to the data using the DPABI toolbox in the DPARSF package (http://rfmri.org/DPARSF accessed on 1 April 2021) [37].

Note that we conducted all analysis in native space for both patients and healthy volunteers in order to avoid misalignment errors related to the gross brain abnormalities that characterize our patient group.

### 2.4. Selection of ROIs

We used four regions of interest (ROIs) to characterize the DMN (medial frontal cortex; PCC; bilateral inferior parietal lobules), in line with previous DCM reports focusing on this network [6,38,39,40]. In addition, we included four ROIs to represent the anterior forebrain mesocircuit (bilateral thalamus; bilateral striatum). To functionally localize each region in each participant, we conducted a seed-based connectivity for each network using the PCC and the striatum as seeds for the DMN and the AFM, respectively. For this, first we identified coordinates for the PCC and bilateral striatum manually for each individual by visual inspection of the T1. To locate the PCC, we moved 6mm left in the x-axis from the AC-PC and located the space between the marginal sulcus and the parieto-occipital sulcus in sagittal view. From here, to identify the center of our sphere in the z plane, we used a reference line crossing the upper boundary of the corpus callosum. This resulted in centring the sphere in the same anatomical subregion that was reported by Fox et al. (2005) [41]. For the striatum, we made note of the coordinates 5 mm above the AC-PC line on the coronal plane, as well as the coordinates of the point at which the superior edge of the dorsal striatum ends, bordered by the lateral ventricle, as recommended elsewhere [42]. After finding the coordinates halfway between these two points, we used the sagittal and axial axes to find the center of the dorsal striatum by inspection, checking that the sphere was going to fully remain within the anatomical boundaries of the region. In addition, we identified coordinates for two spheres to be located in the cerebrospinal fluid (CSF) and white matter (WM) for their time series to be used later as regressors of no interest. We used the posterior horn of the right lateral ventricle for the CSF, and the area anterior to the anterior horn of the right lateral ventricle for WM, ensuring 5 mm spheres would fit within their respective anatomical boundaries.

After this, we used the MarsBaR SPM toolbox [43] to generate spheres centered on the above coordinates for each individual: we set the radius at 8mm for the PCC, 3 mm for right and left striatum, and 5 mm for CSF and WM, to accommodate for anatomical differences in the size of each structure. We then ran seed-based connectivity analysis on SPM independently for the PCC and bilateral striatum using CSF and WM as nuisance regressors. For each individual, we then located each region of interest (MPFC, and IPL for the DMN and thalamus for the AFM) visually and saved the coordinates of the nearest local maxima to use in the extraction of time series for the DCM. Note that we used connectivity with PCC to identify the DMN regions, and connectivity with striatum to identify the AFM. As visual reference, for the medial prefrontal cortex, we placed the cursor in the middle of the frontal medial cortex in the area between the paracingulate gyrus and the frontal pole (as per Harvard-Oxford Cortical Structural Atlas); to locate the IPL, we placed the cursor in the middle of the angular gyrus, located between the lateral occipital cortex and the supramarginal gyrus. Finally, to locate the coordinates for the left and right thalamus, we simply placed the crosshairs in the center of each before selecting the nearest local maximum. Note that, due to the level of subcortical damage that characterizes PDOC patients, and the spatial resolution of our data, it was not possible to accurately identify the globus pallidus in most patients and we will not include this region in our analyses.

For all regions, we visually inspected the location of the nearest local maxima selected in the above step to ensure the area was still appropriate anatomically. If no local maxima was present in the appropriate anatomical region, we used anatomical coordinates (based on our visual inspection as described above) to extract the timeseries for the DCM. Figure 1 displays the ROIs used in the DCM for a representative healthy volunteer overlaid onto their T1 image.

### 2.5. General Linear Model

Note that the above ROI selection was done independently of the analyses used to draw inferences about effective connectivity. For this, we ran a second linear model to extract the time series to be used during dynamic causal modelling. This included 6 rigid body realignment parameters to account for head motion in our general linear model, as well as white matter and cerebral spinal fluid mean signals as nuisance regressors. We discarded volumes with levels of motion above 3 mm and 3 degrees for patients. No volumes were above 2 mm translation and 2° radius for any of the healthy volunteers.

### 2.6. Dynamic Causal Modelling

We performed bilinear, one-state spectral DCM with the DCM12 routine implemented on SPM12 on MATLAB version R2014b. Spectral DCM operates in the frequency domain rather than time and is therefore better suited to resting state data analysis [44] (see [44] for more details). We first extracted the time series using the ROIs with spheres centred on the individual coordinates previously generated for PCC, MPFC, left and right IPL (all 8 mm radius), bilateral striatum and thalamus (all 3 mm radius). The difference in radius-reflected differences in anatomical size for cortical vs subcortical regions. We specified our model space based on a fully and reciprocally connected model, resulting in a total of 64 effective connections (i.e., the A-matrix) produced and estimated for each model.

### 2.7. Parameter Estimations—Parametric Empirical Bayes (PEB), Bayesian Model Reduction (BMR) and Bayesian Model Averaging (BMA)

After the model specification, we took the first level individual (within-subjects) estimated fully connected DCMs to a second level (between-subjects) analysis using parametric empirical Bayes (PEB) [45]. PEB is a hierarchical approach to model how individual subjects’ connections relate to the group level by using the individual DCMs as priors (first level) to constrain the variables in the Bayesian linear regression model (second level) [46]. This method allows individual variability in connection strengths to then influence the group (second level) analysis. After group level connection strengths (parameters) have been estimated, hypotheses are tested by comparing evidence for different variations of these parameters in a process known as Bayesian model comparison.

In order to assess the differences in connectivity strength between patients and healthy controls we first built a PEB model including the average connectivity in the healthy group (as baseline), the intercept of differences between healthy controls (0) and patients (1), and the mean centered age as non-interest regressor (to account for the effect of differences in age between the groups, *p* = 0.002).

We subsequently ran a second PEB model including only PDOC patients to assess the canonical connectivity in this group, also using mean centered age as covariate. This was used to extract connectivity parameters to conduct correlations with clinical variables (see CRS-R below).

For both PEBs, after fitting each model we used Bayesian model comparison to prune away parameters that are not contributing to the model evidence. This was achieved by performing a search over nested models where parameters are removed from the full model hierarchically. We then used Bayesian Model Reduction (BMR) to prune connections from the full model until there are no more improvements in model-evidence. This approach uses the group parameters as empirical priors to re-estimate the individual parameters, and in doing so, limits the effects of outliers [47]. Parameters from the best models following BMR were then taken, weighted by their model evidence, and combined using Bayesian model averaging (BMA). To define statistical significance, we applied a threshold of a posterior probability > 0.95 (strong evidence) for free energy for each connection (i.e., comparing the evidence for all models in which the particular connection is on with those where the connection is switched off). For more information on BMR and BMA, see Friston et al., 2016 and Zeidman et al., 2019 [45,46].

### 2.8. Correlations with CRS-R

To investigate any relationship between the effective connectivity and level of consciousness in the PDOC group, we extracted re-estimated parameters (Ep.A values) from the PDOC PEB for each connection that was significantly different in the comparison between PDOC and HC and correlated them with CRS-R on the day of scan and maximum CRS-R on the week of scan using the Kendall’s Rank Correlation Coefficient (both frequentist and Bayesian implementations) in JASP [48]. It should be noted that CRS-R score on the day of scan was unavailable for 3 of the patients (see Table 1). We will define outliers as any data point equal or greater than 2 standard deviations from the mean. For this analysis, we converted self-connection parameters to Hz using y = −0.5 ∗ exp(x), where x is the log scaling parameter (Ep.A value), −0.5 Hz is the prior, and y is the self-connections strength in Hz. We set statistical significance at two-tailed *p* < 0.05. For the Bayesian analysis, we used a Jeffrey-Zellner-Siow Bayes factor (JZS-BF10) to contrast the strength of the evidence for models supporting a relationship between the variables versus the null. A JZS-BF10 between 0.33 and 3 is considered to be weak/anecdotal evidence for an effect; 3–10: substantial evidence; 10–100: strong evidence; >100: very strong evidence. To consider any support for the null we also calculated JZS-BF01.

## 3. Results

### 3.1. Effective Connectivity

Figure 2 shows the effective connectivity matrix depicting the mean for healthy controls (A) as well as the differences between healthy controls and patients (B) in the reduced model: i.e., only for those parameters with evidence above our threshold of 0.95 posterior probability; the lower half of the figure displays the differences between groups as a schematic for DMN (C) and AFM (D) separately, and well as for the connections between both networks (E).

As our main focus is the difference between groups, we will only make a few remarks regarding the pattern of effective connectivity observed in healthy controls: the PCC has a clear excitatory role over all other regions of the DMN; each striatum excites the ipsilateral thalamus; the AFM is mostly inhibitory towards the DMN, with a marked lateral component (left thalamus and right striatum); and the DMN seem to modulate the AFM mostly through frontal (MPFC) rather than parietal (PCC) medial areas.

In terms of group differences, within the DMN we observe disruptions only in self-connections of medial areas, with PDOC patients having decreased self-inhibition of the PCC and increased self-inhibition of the MPFC compared to healthy controls. Note that parameters for self-connections are log scale parameters and thus negative differences (blue cells in Figure 2B) reflect less self-inhibition in patients. In contrast, parameters for the between region connections reflect differences in the connection strength across groups, and thus negative values (blue cells) represent reduced coupling in PDOC. In the AFM, PDOC patients show less self-inhibition in both striata, as well as the left thalamus. In turn, most connections between AFM regions also show reduced coupling in PDOC, resulting in overall reductions in connectivity. We did not identify any connection with increased coupling in PDOC for the AFM. Crucially, PDOC patients showed decreased coupling from both striata to the ipsilateral thalamus, which would result in an increased inhibitory tone. The reduced thalamic self-inhibition in the left hemisphere would in turn suggest that the left thalamus is more readily affected by the inhibitory striatal input.

In terms of extrinsic (between networks) connections, PDOC patients showed increased coupling, reflecting an overall reduction in inhibition from the AFM to the DMN. Specifically, this affected afferent connections from left thalamus to PCC and right IPL, left striatum to MPFC, and right striatum to both IPLs. In contrast, the connection from right striatum to MPFC showed a very weak negative value. However, upon further investigation, its 95% confidence interval included zero values (Appendix A) and therefore it is not possible to determine the sign of this effect (even when this connection contributes to the model evidence, albeit weakly).

Finally, for the connections from DMN to AFM, PDOC patients showed a decoupling of afferent connections from midline regions. This had a laterality effect, with the PCC showing reduced coupling with both right thalamus and right striatum, and MPFC showing reduced coupling with the equivalent regions (thalamus and striatum) in the left hemisphere. Similarly, the lateral parietal areas significantly lost their natural inhibitory tone to the right thalamus (displaying an increased coupling) in PDOC. In addition, the left and right IPL showed increased and decreased coupling with the left thalamus respectively, but we did not find these connections to contribute to the healthy controls model (Figure 2A) and therefore the differences in PDOC are not straightforward to interpret.

For clarity, Figure 3 shows the strengths of the connections that showed differences between groups.

### 3.2. CRS-R and Effective Connectivity

We found significant correlations with CRS-R score on the day of scan (n = 13) for the left thalamus self-connection (r = −0.477, *p* = 0.029, JZS-BF10 = 3.630) and the connection between right thalamus to left thalamus (r = −0.450, *p* = 0.040, JZS-BF10 = 2.804). Both were, however, driven by outliers, and when these were removed, they both were no longer significant (left thalamus: n = 12, r = 0.375, *p* = 0.105, JZS-BF10 = 1.331; right thalamus to left thalamus: n = 11, r = −0.350, *p* = 0.150, JZS-BF10 = 1.032). See Appendix A. We found no significant correlations for the maximum CRS-R on the week of scan. In addition, when outliers were removed, we found evidence for the lack of a relationship between maximum CRS-R and four connections: MPFC to left thalamus (JZS-BF01 = 3.045), RIPL to right thalamus (JZS-BF01 = 3.154); left thalamus to left striatum (JZS-BF01 = 3.14), and left striatum to MPFC (JZS-BF01 = 3.1). See Appendix A.

## 4. Discussion

In this study, we provide the first report of disruptions in effective connectivity within and between the DMN and AFM in PDOC patients using spectral DCM of fMRI. We show marked disruptions in coupling mostly affecting the AFM and key connections from and to the DMN.

### 4.1. The AFM and Extrinsic Connections to the DMN

PDOC patients showed extensive decoupling between most regions of the AFM, alongside reduced self-inhibition across all but the right thalamus (no differences) compared to healthy controls. As expected, healthy controls showed bi-directional excitation between the striatum and the thalamus, in line with the well-known canonical interactions between these regions at the neuronal level [20]. The mesocircuit hypothesis posits that, in PDOC, the striatum fails to inhibit the globus pallidus, which in turn becomes hyperactive and excessively inhibits the thalamus. This catalyzes a widespread downregulation of the anterior forebrain and its cortical projections [20]. In the current study, we did not have the spatial resolution to reliably identify the globus pallidus as a region of interest to include in our DCM models, and we included a direct connection between striatum and thalamus instead. However, DCM does not assume direct anatomo-functional connections between the regions and thus our effects for the connection from striatum to thalamus are likely to be mediated by the globus pallidus. As predicted, PDOC patients showed reduced coupling from each striatum to the ipsilateral thalamus as compared to controls. The left thalamus was also less self-inhibited, which made it more vulnerable to this reduced excitation from the striatum (or increased inhibition from the globus pallidus). In turn, the left thalamus showed reduced coupling with the striatum, which was also more responsive to external inputs, further contributing to a reduced excitatory output back to thalamus. Overall, our results are in line with previous research showing a reversal in the resting metabolic profiles of the globus pallidus and thalamus in PDOC patients compared to controls using PET (specifically, an increase in globus pallidus metabolism alongside a reduction in central thalamus metabolism in patients) [16]. Therefore, we provide further support for the mesocircuit hypothesis to explain forebrain disfunctions in PDOC [20]. In addition, our results suggest a hemispheric asymmetry, with the left hemisphere more strongly contributing to the differences between PDOC and controls.

Although the aim of our study was to investigate the neural mechanisms underlying PDOC, our results may also contribute to understanding the success (or otherwise) of pharmacological and stimulation therapies. For example, a number of studies have observed clinical improvements in PDOC following administration of amantadine [49]. The proposed mechanism of action is an excitatory modulation of the mesocircuit (and its cortical projections) via targeting the striatum [49]. Our findings confirmed a reduced excitatory coupling from striatum to thalamus in PDOC, providing further support to the use of pharmacological agents that can restore this coupling.

Against our prediction, the AFM showed a widespread inhibitory tone towards the DMN in healthy controls, and a reduction of this inhibition in PDOC. The AFM typically exerts an excitatory role over fronto-parietal networks, which is thought to be reduced in PDOC [50]. While the specific cortical networks involved in this subcortico-cortical disbalance in PDOC are not fully defined, we have previously argued that the DMN is likely to have a central role, on the basis of its strong structural connectivity with the AFM and its widely reported functional and metabolic impairments in this patient group [8]. Our current results confirm an important relationship between both networks but, interestingly, this was in an unexpected direction. It is well known that the DMN is typically deactivated during tasks [41], and anti-correlates with dorsal fronto-parietal networks associated with external awareness and high order cognitive functions and executive control [51]. It is thus possible that the AFM exerts a different tone over the DMN as compared to its anti-correlated dorsal networks. The study of such dorsal networks was beyond our scope and therefore this conclusion remains necessarily speculative, but our findings could be suggesting a role for the thalamus, and the rest of the AFM, in the modulation of excitation and inhibition between anticorrelated networks, which would be altered in PDOC and might result in an inability to filter internal and external stimuli. In either case, our results suggest that the lack of awareness in PDOC may be underpinned by lack of inhibition to the DMN rather than lack of excitation. Interestingly, previous research in severe traumatic brain injury has revealed increased functional connectivity in the DMN compared to controls. While this study focused on patients who had recovered consciousness, their findings highlight the complex relationship between brain injury and observed disruptions to brain function, which can be translated in both hypo- and hyper-connectivity [52].

The relationship between AFM and DMN once again showed a hemispheric asymmetry, with the left thalamus and the right striatum driving the effects for their respective hemisphere. Specifically, patients had increased coupling from the left thalamus to the PCC and right IPL, as well as from the right striatum to both IPLs. In addition, the left striatum showed increased coupling with MPFC. Previous research suggests that structural differences are more prominent in the left hemisphere in PDOC [53,54], and atrophy in the left thalamus predicts functional outcome in post-traumatic brain injury survivors [55]. Furthermore, the metabolic integrity of the left hemisphere better predicts clinical diagnosis (vegetative versus minimally conscious state) [56]. In a recent ultrasound stimulation study on three chronic PDOC patients, stimulation of the left thalamus elicited some promising improvements in CRS-R scores [57]. Together, these studies provide evidence for a laterality effect in the neural bases of PDOC, although the reasons behind the higher vulnerability of structures in the left hemisphere remains unclear.

### 4.2. The DMN and Extrinsic Connections to the AFM

PDOC patients were characterized by differences in the self-inhibitory tone of the two midline regions of the DMN (PCC and MPFC), but no differences in any intrinsic (region to region) connections. Specifically, they showed reduced self-inhibition of the PCC and increased self-inhibition in MPFC.

The observed reduced PCC self-inhibition would leave this region more susceptible to afferent inputs from other regions, which in our case were limited to the above discussed increased coupling from the left thalamus. Therefore, PCC self-inhibition would make the DMN more vulnerable to the effects of the AFM modulations. This result is in line with an earlier report investigating effective connectivity changes in the DMN of PDOC patients [6]. They observed reduced self-inhibition of the PCC in vegetative state compared with minimally conscious state patients, which correlated with level of consciousness [6]. Similarly, a PET study reported reduced effective connectivity (based on reduced glucose metabolism) of the PCC in vegetative state patients compared with controls [15]. The PCC is widely regarded as a central hub for functional integration of information within the DMN and across networks [26,27,28], and it has been noted this pathway could play a central role in consciousness [5,50]. We have previously reported specific structural damage to all white matter connections, connecting the PCC with the rest of the DMN and the AFM in PDOC [5,8] and more broadly, damage to the PCC area consistently appears as crucial to understanding the neural basis of PDOC [58]. Interestingly, two recent transcranial direct current stimulation (tDCS) studies attempted to target the PCC with high definition-tDCS and both found clinical improvements in minimally conscious and vegetative state patients [59,60].

Leaving differences between PDOC and controls aside, previous research from healthy populations reported inconsistent results on the role and tone of the influence of the PCC to the rest of the DMN at rest. Some studies observed all driving influences from the PCC were negative [6,40], whilst others found the PCC exerts an excitatory tone over the network [61,62]. In our sample, we found support for the latter, with the PCC displaying excitatory output to all remaining regions of the DMN. This tone did not appear to differ in PDOC.

In contrast, we observed a marked increase in the self-inhibition of the MPFC in patients. This would lead to the MPFC being less influenced by inputs from within the DMN (specifically the PCC and LIPL) and the AFM (bilateral thalamus and left striatum). This is in line with previous research using graph theory on functional connectivity that found enhanced MPFC connectivity in PDOC [63]. Interestingly, their effects were driven specifically by minimally conscious patients, with no changes between vegetative state and controls. While our sample size does not allow for disaggregated comparisons across diagnostic categories, our cohort includes a large proportion of vegetative state patients (11/16), suggesting that our differences are not likely to be driven by minimally conscious patients alone. In either case, our results agree in suggesting more marked disruptions in posterior regions of the DMN in PDOC [58].

Beyond the above discussed changes in self-inhibition, we did not identify any differences in the region-to-region connections of the DMN in PDOC. This is in contrast with previous studies reporting differences in functional connectivity between PDOC and healthy controls, many of which consistently reported disconnections between midline DMN regions [2,56,64,65]. These typically correlate with level of consciousness (based on CRS-R score) and outcome [66]. It is worth highlighting here that functional and effective connectivity tap into very different underlying mechanisms. While functional connectivity analyses identify areas with correlated activity (time series) across time, effective connectivity investigates the causal influence one region exerts over another [67]. Our findings thus suggest that previously reported disruptions in functional connectivity across the DMN in PDOC do not necessarily translate into changes in how they influence or depend on each other.

To our knowledge, only one previous study has looked at effective connectivity withing the DMN in PDOC [6]. The authors reported reduced inhibitory coupling from PCC to MPFC and LIPL in both vegetative and minimally conscious patients [6], which we did not see in our study. It is possible our inclusion of the AFM in our model is explaining this discrepancy and some of the effects in [6] were driven by indirect connections to subcortical regions. Additionally, contrary to Crone and colleagues, we found no differences between healthy controls and PDOC for connectivity from MPFC to either IPL. Interestingly, the authors reported reduced excitation in both connections in vegetative state patients only, while minimally conscious patients did not significantly differ from controls. It is therefore possible that the discrepancy with our results may be due to our minimally conscious patients masking potential differences existing in the vegetative state group. As discussed above, our sample size did not allow for disaggregation of diagnostic categories in our analyses. Future research with larger samples is needed to disentangle these potential differences. We also note that we conducted our analyses in native space with manually defined masks. While Crone and colleagues embedded robust steps in their analyses to account for potential confounds due to brain abnormalities, we cannot rule out that differences in the pipeline are explaining the discrepancy in our results. This highlights the need for further research to investigate these discrepancies.

The influence of cortical networks on the AFM is much less understood. Here, we saw a decoupling from the PCC to right thalamus and right striatum in PDOC, as well as from the MPFC to left thalamus and left striatum. This suggests a lack of excitation, which would further contribute to the downregulation of the AFM. Interestingly, we also identified a laterality effect with PCC and MPFC modulating the right and left hemispheres, respectively.

### 4.3. Clinical Severity

We were unable to identify a relationship between the differences in effective connectivity observed between healthy controls and PDOC and clinical severity as indexed by CRS-R. We note that our sample size was small and thus our correlations may be underpowered, specifically for CRS-R on the day of scan where the scores for three patients were unavailable. Indeed, our Bayesian analyses confirmed lack of support for a relationship or a lack of for most connections. Our small sample size also precluded us from undertaking comparisons between vegetative state and minimally conscious state. Therefore, while we can conclude that the above discussed differences in effective connectivity contribute to explaining the neural bases of disorders of consciousness, we cannot make strong arguments about the specific relationship between these two different levels of awareness in PDOC.

## 5. Conclusions

In summary, the current study aimed to establish whether previously reported changes in the DMN in PDOC are caused by lack of excitation from the AFM and the role of the PCC in modulating these effects. PDOC patients showed an overall downregulation of the AFM, likely caused by an increased inhibitory tone of the striatum over the thalamus, providing support for the anterior mesocircuit model. Moreover, in PDOC, The AFM modulated the DMN mostly through posterior areas, including the PCC and IPL. However, instead of identifying a loss of excitation from AFM to DMN in PDOC, we found the AFM has an inhibitory tone over the DMN in the healthy brain at rest and this is disrupted in PDOC. Overall, our results suggest that complex disruptions in the interplay between DMN and AFM characterize PDOC.

## Figures and Tables

**Figure 1 brainsci-11-00749-f001:**
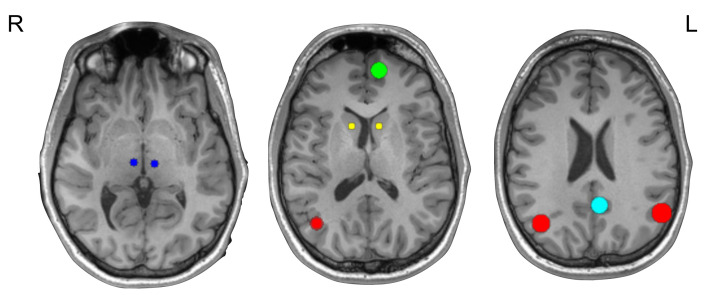
Regions of interest for DCM analyses, on the T1 of a representative healthy control and. Red, IPL; green, MPFC; light blue, PCC; dark blue, thalamus; yellow, striatum; L, left hemisphere; R, right hemisphere; HC, healthy control.

**Figure 2 brainsci-11-00749-f002:**
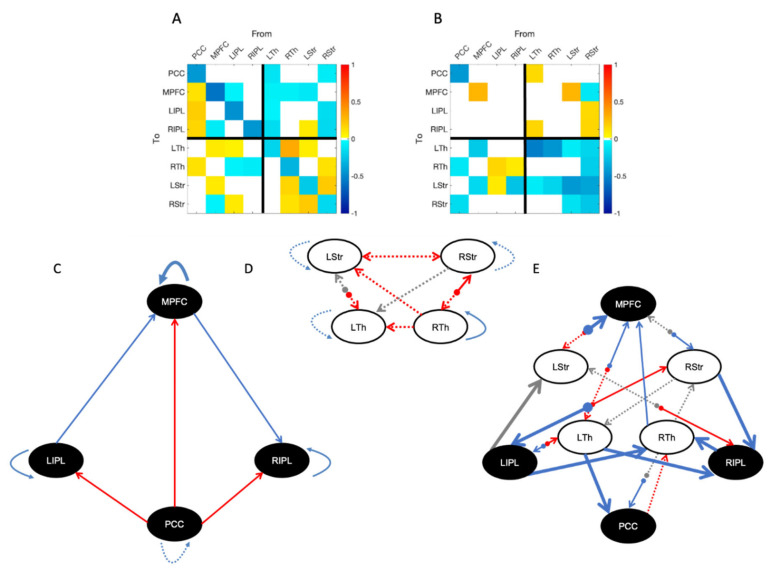
Group mean effective connectivity in healthy controls (**A**,**C**–**E**) and differences with PDOC patients (**B**–**E**). We only display connections present in the reduced model (>95% posterior probability for free energy). The top two panels, (**A**,**B**), show the mean parameter strength (Ep.A) for the healthy controls mean and the difference between them and PDOC, respectively. Note that self-connections are always inhibitory meaning a positive self-connection parameter in B indicates that PDOC have stronger self-inhibition compared to healthy controls (negative being the opposite). (**C**,**D**) show schematic representations of the results in the matrix in **B** separately for the DMN (**C**), the anterior forebrain mesocircuit (AFM) (**D**), and the extrinsic connections between them (**E**). Note that this is only to facilitate visualization, but all regions were part of the model space in our analysis. The color of the line refers to its tone in the healthy control mean: red lines represent excitatory connections, and blue represent inhibitory. Grey lines show connections that did not contribute to the healthy control mean model but showed differences between them and PDOC. The format of the line represents the differences between groups: a dashed line represents reduced coupling in PDOC and a thick line shows stronger coupling in PDOC. For self-connections, dashed lines represent reduced self-inhibition and thicker lines increased self-inhibition. PCC, posterior cingulate cortex/precuneus; MPFC, medial prefrontal cortex; LIPL, left inferior parietal lobule; RIPL, right inferior parietal lobule; LTh, left thalamus; RTh, right thalamus; LStr, left striatum; RStr, right striatum.

**Figure 3 brainsci-11-00749-f003:**
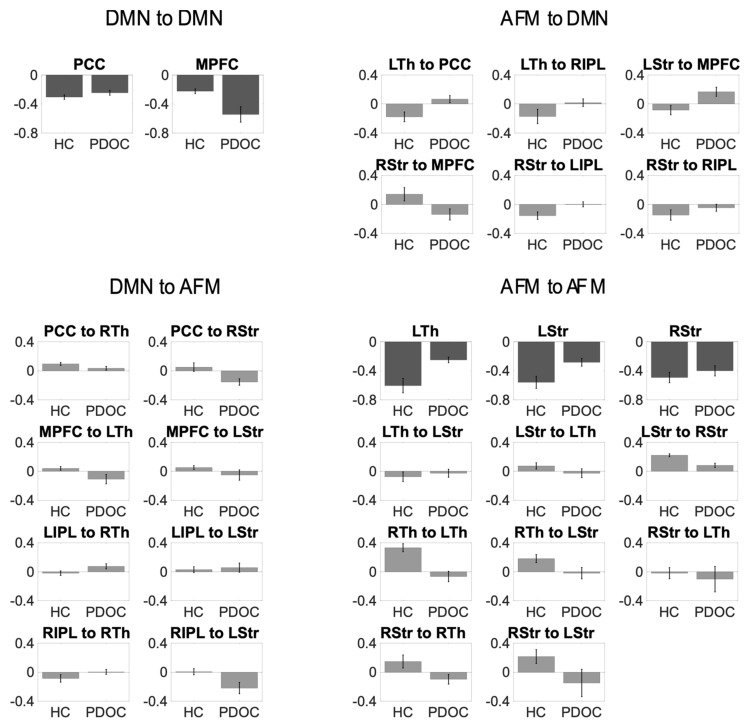
Mean connection strength (Ep.A) for the connections contributing to the difference between groups. Error bars represent standard error. Self-connections appear in darker grey and have been converted into Hz. PCC, posterior cingulate cortex/precuneus; MPFC, medial prefrontal cortex; LIPL, left inferior parietal lobule; RIPL, right inferior parietal lobule; LTh, left thalamus; RTh, right thalamus; LStr, left striatum; RStr, right striatum.

**Table 1 brainsci-11-00749-t001:** Clinical information.

Patient No.	Sex	Age (y)	Interval Since Ictus (Months)	Etiology	Diagnosis	CRS-R on Day of Scan	Max CRS-R on Week of Scan	Auditory Function	Visual Function	Motor Function	Oromotor/ Verbal	Communication	Arousal	Number of CRS-R Scores Taken on Week of Scan
1	Male	38	150	Traumatic	VS	6	6	1	0	2	1	0	2	1
2	Male	33	176	Non-Traumatic	MCS	9	10	1	3	2	1	0	2–3	5
3	Male	27	88	Traumatic	VS	7	7	1–2	0–1	2	1	0	1–2	5
4	Female	44	245	Traumatic	VS	3	5	0–1	0–1	0–2	0–1	0	0–2	4
5	Female	46	230	Non-Traumatic	MCS	-	10	1–2	3	2	1	0	1–2	2
6	Male	57	37	Non-Traumatic	VS	-	6	1	0–1	0–2	1	0	1–2	3
7	Male	27	36	Non-Traumatic	MCS	-	13	3	3	0–2	2	0	3	3
8	Female	20	67	Non-Traumatic	VS	6	8	1–2	0–1	1–2	1	0	2	4
9	Female	35	24	Non-Traumatic	VS	5	5	0–1	0	0–2	1	0	1–2	3
10	Male	19	2	Non-Traumatic	VS	6–7	7	2	1	1	0–1	0	2	3
11	Female	25	68	Traumatic	MCS	8	9	1–2	3	1	1	0	2	5
12	Female	43	55	Non-Traumatic	VS	5	7	1	0–1	2	1	0	1–2	3
13	Male	20	48	Non-Traumatic	VS	5	6	1	0–1	0–2	1	0	1–2	3
14	Female	51	11	Non-Traumatic	VS	4	4	3–4	1	0–1	0–1	0	1	5
15	Female	52	78	Non-Traumatic	VS	6	6	1	0	1–2	1	0	1–2	4
16	Male	40	38	Traumatic	MCS	7	7	1	0–3	0–1	0–1	0	1–2	5

CRS-R, coma recovery scale-revised; VS, vegetative state; MCS, minimally conscious state.

## Data Availability

The data presented in this study are not available due to ethical restrictions.

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
