# Peer review of "Disruptions in Effective Connectivity within and between Default Mode Network and Anterior Forebrain Mesocircuit in Prolonged Disorders of Consciousness"

_brainsci, 2021, doi:10.3390/brainsci11060749_

Round 1

Reviewer 1 Report

In  the original manuscript entitled :Disruptions in effective connectivity within and between default mode network and anterior forebrain mesocircuit in prolonged disorders of consciousness, Dr. Coulborn and collaborators evaluated using rs-fMRI the neural base of prolonged DOC by means of spectral dynamic causal modelling and parametric empirical Bayes. Specifically, they tested the hypothesis that cortical changes in the default mode network (DMN) in prolonged DOC are caused by 1- lack of excitation from the thalamus (anterior forebrain mesocircuit, AFM); and, 2- that the posterior cingulate/precuneus complex (PCC) modulates (negatively?) this lack of excitation from the thalamus.

Although the reported dataset comes from local clinical and neuroimaging repository data from previous published manuscript, authors implemented a different statistical approach to test the proposed hypothesis. Overall, the paper is well written and displays enough background to support the theoretical and methodological framework. However, there are some aspects that would require a major revision as stated below.

Major issues:

Methods and results: the described population consisted of 18 prolonged DOC with two cases discarded due to motion artifact (n=16). There is no mention of outliers and/or criteria to define them at any part of the manuscript but on the very last paragraph of results [i.e., 3.2. CRS-R and effective connectivity, “We found significant correlations with CRS-R score on the day for the left thalamus self-connection (r = -0.477, p = 0.029, BF10 = 3.630) and the connection between right thalamus to left thalamus (r = -0.450, p = 0.040, BF10 = 2.804). Both were however driven by outliers, and when these were removed, they both were no longer significant (left thala-mus: r = 0.375, p = 0.105, BF10 = 1.331; right thalamus to left thalamus: r = -0.350, p = 0.150, BF10 = 1.032). See Supplementary Figure 2”]. However, figure S2 (n=13) depicts a single outlier and data set lacks of 3 cases from the total cases reported (n=16) at first and second level analysis. Are these 3 cases outliers as well? If yes, what were the arguments to consider them outliers? Why were they consider outliers for this stage of the analysis but not for 1st and 2nd hierarchical stats? Given that clinical and neuroimaging information referred in the manuscript highlights for the AFM the central role of the thalamus in controlling the DMN outflow (i.e., CRS-R, rs-fMRI and FDG-PET), it would be essential to clarify this issue point by point in methods, results and discussion sections.

Minor issues:

  • Table 1 is out of range and does not allow full interpretation of CRS-R data.
  • On page 5, point 2.7. Parameter estimations - parametric empirical Bayes (PEB), Bayesian model reduction (BMR) and Bayesian model averaging (BMA), first paragraph, please correct the extra-space in the sentence.
  • On page 6, point 2.7. Parameter estimations - parametric empirical Bayes (PEB), Bayesian model reduction (BMR) and Bayesian model averaging (BMA), last paragraph: please clarify BF, JSF-BF (same on results).
  • Supplementary Figure 1, there is only one variable described (i.e., the problematic). It would help knowing at least the name of the ones that were not described but tested.

Reviewer 2 Report

Interesting work. I have a couple of suggestions:

- the clinical description of the patients is not sufficient (e.g., treatment? rehab setting, general brain MRI pattern).

- Table 1 is not completely visible in the pdf file

- a rehab/management perspective of the present findings could be valuable.

Reviewer 3 Report

The Authors show disruptions in effective connectivity within and between the Default Mode Network (DMN) and anterior forebrain mesocircuit (AFM) in patients with prolonged disorders of consciousness (PDOC).

- The main problem is the fact that vegetative state and minimal consciousness are not separated. This is indeed mentioned by the Authors in Discussion, but this could possibly explain some non significant results or incongruences with previous literature, therefore the Authors should elaborate on this more extensively in Discussion, and possibly introduce the subject already in the Introduction

- Methods: please check acquisition parameters: in the first set, number of slices and FOV are missing, and something is likely wrong in the matrix (it would make a FOV > 1 meter); in the second set, in-plane resolution, slice number and thickness are missing.

- Please check for typos, e.g.: “hipper-connectivity”; “assimetry”; “PCC excerpts an excitatory tone”, etc; also, the sentence “While functional connectivity observes areas which neural activity...” is meaningless as it is written.

Round 2

Reviewer 1 Report

Good work. Congratulations.